# Research on Parameter Optimization Method of Sliding Mode Controller for the Grid-Connected Composite Device Based on IMFO Algorithm

**DOI:** 10.3390/s23010149

**Published:** 2022-12-23

**Authors:** Ji Sun, Jiajun Liu, Miao Miao, Haokun Lin

**Affiliations:** School of Electrical Engineering, Xi’an University of Technology, Xi’an 710048, China

**Keywords:** back-to-back MMC-HVDC, grid-connected composite device, sliding mode control, moth-flame optimization algorithm, Python–PSCAD joint simulation

## Abstract

In order to make the grid-connected composite device (GCCD) controller meet the requirements of different operating modes and complex working conditions of power grid, this paper proposes to introduce sliding mode control (SMC) into GCCD controller. Firstly, the mathematical model of MMC converter is established, and the sliding mode controller is designed based on the SMC principle. Then, aiming at the problems of complex controller structure and difficult parameter tuning in multiple modes of the GCCD, this paper proposes a controller parameter optimization method based on improved Month Flame optimization (IMFO) algorithm. This method improves the *MFO* algorithm by introducing good point set (GPS) initialization and Levy flight strategy, which accelerates the convergence speed of the algorithm while avoiding falling into local optimization, and realizes the optimization of converter controller parameters. Under a variety of standard test functions, the advantages of the proposed IMFO algorithm are verified by comparing it with the traditional algorithm. Finally, in order to realize the automatic tuning of control parameters, the Python–PSCAD joint simulation method is studied and implemented. Taking the comprehensive integral of time and absolute error (*CITAE*) index as the objective function, the parameters of the sliding mode controller are optimized. The simulation results show that the controller parameters optimized by the IMFO algorithm can make the GCCD have better dynamic performance.

## 1. Introduction

With the intelligent and automatic development of the interconnected power grid, the research on reliable and effective new grid connection mode is crucial [1]. Some scholars put forward a GCCD based on back-to-back VSC-HVDC. By transferring active and reactive power between systems on both sides to be paralleled, the frequency and voltage of both sides to be paralleled are adjusted to achieve synchronous paralleling between power grids. This method changes the current situation that the traditional synchronous paralleling method completely relies on manual operation, which involves a wide range of operations and is difficult, and the grid connection speed is slow, and the success rate is low [2,3].

In order to apply the GCCD to the actual project, the Ref. [4] puts forward the calculation method of the capacity of the device, as well as the selection principle and calculation method of parameters, which provides a basis for the actual project application, design and economic analysis of the device. In order to improve the utilization rate of the device, the team proposed to convert the device to the FACTS device in the subsequent study, and defined the conversion topology and control strategy of grid connection, tie line power flow control and splitting modes [5]. In order to adapt to the low inertia characteristics of the new power system, the Ref. [6] studied the control strategy of the device applied to the low inertia power system. The GCCD based on back-to-back MMC-HVDC has a complex topology and multi-link control strategy, so it is necessary to select a control strategy with good control performance. The advantages and disadvantages of current common control strategies are shown in Table 1.

During mode conversion, the dynamic performance of the GCCD is the most important indicator. Through analyzing the advantages and disadvantages of common control strategies, the SMC is selected as the control strategy of the GCCD. Due to the large number of the SMC parameters and the multiple modes of the GCCD, it will be more difficult to set the parameters during mode switching, and the conventional empirical trial and error method will gradually fail to meet the requirements [11,12]. In order to ensure the fast and smooth switching of device mode, the research on automatic tuning of converter control parameters is of great significance.

At present, most of the tuning methods of converter control parameters use intelligent algorithms to optimize, taking the output effect as the optimization objective and the converter control parameters as variables for multi-objective optimization. Common intelligent optimization algorithms include GA algorithm, PSO algorithm, etc. [13]. In order to improve the efficiency of multi-objective optimization, scholars at home and abroad are also constantly improving intelligent algorithms.

Mirjalili S team proposed the *MFO* algorithm, and compared it with other famous natural heuristic algorithms on 29 benchmarks and 7 practical engineering problems. The comparison results show that the algorithm is feasible and superior [14]. In view of the difficulty in tuning the PI control parameters of wind turbine generators, some scholars used the *MFO* algorithm to optimize the controller parameters, and the experiment proved that the control effect of the PI controller after parameter optimization was significantly improved [15]. In view of the difficulty of PSS parameter coordination in multiple operation modes, some scholars proposed to apply the *MFO* algorithm to PSS parameter coordination optimization. The simulation results show that the dynamic stability of the system is effectively improved after the application of this method [16]. For the optimal power flow problem of power system, some scholars have proposed an optimization solution scheme using *MFO* algorithm. The results of an example show that using *MFO* algorithm to solve the optimal power flow problem has the advantages of faster convergence speed, higher search accuracy, and strong robustness [17]. The above research shows that the *MFO* algorithm has been widely used in the optimal solution search of power system. However, the algorithm still has the problems of large global search complexity and local search easy to fall into local optimization in practical application.

In order to ensure the optimization effect of the algorithm on the controller parameters of the GCCD based on back-to-back MMC-HVDC, this paper improves the *MFO* algorithm from the perspective of both global search and local search, so that it can better optimize the sliding mode controller parameters of the GCCD. The controller can adaptively adjust parameters according to different working conditions to meet the requirements of different modes of the device and complex working conditions of the power grid.

The main contributions of this paper are as follows:

(1)The mathematical model of MMC converter is established, and the controller of GCCD is designed according to the SMC principle, so that GCCD can meet the requirements of different operation modes and complex working conditions of the power grid.(2)By improving the *MFO* algorithm through GPS initialization and Levy flight strategy, this paper proposes an optimization method of converter control parameters based on IMFO algorithm. Through comparison and verification, the above improvements can effectively improve the convergence speed of the algorithm, avoid falling into local optimization, and significantly improve the optimization performance of the algorithm.(3)In order to optimize the control parameters of the converter accurately, this paper proposes a parameter optimization method of sliding mode controller based on IMFO algorithm. Through Automation Library as a link, automatic parameter tuning is realized in Python-PSCAD joint simulation. By comparing the step response performance of non-optimization, *MFO* and IMFO, it is verified that the proposed method can effectively improve the GCCD control performance.

## 2. Sliding Mode Controller of the GCCD Based on Back-to-Back MMC-HVDC

### 2.1. The GCCD Based on Back-to-Back MMC-HVDC

Jiajun Liu’s team proposed the basic principle and control strategy of the GCCD, which can work in the grid-connected mode and the tie line power flow control mode. Its topology and mode switching process are shown in Figure 1 [5,6].

When the device is working in the grid connection mode, it will be put between the two systems to be paralleled. At this time, the device will quickly transfer the active power of the high-frequency side system to the low-frequency side system through the control strategy, so as to reduce the frequency difference on both sides. At the same time, the device injects capacitive reactive power to the parallel points with lower voltage and absorbs reactive power from the parallel points with higher voltage to adjust the voltage amplitude and phase angle of the parallel points. Meet grid connection conditions through power transmission to achieve rapid grid connection. After the device exits the grid connection mode, in order to improve the device utilization, the device can be switched to the tie line power flow control mode. At this time, the structure is the same as UPFC, which can adjust the tie line power flow and improve the power supply capacity.

With the development of power electronics technology and the continuous improvement of the voltage level in the use scenario, the converter has also been upgraded from the original VSC type to the MMC type. The MMC three-phase topology and sub-module structure are shown in Figure 2. The converter sub-module of the GCCD in this paper adopts a half bridge sub-module topology, which is simple in topology, requires less devices and has low overall loss. For MMC with level above 21, the nearest level modulation method(NLM) is often used. The principle of this modulation method is simple, and the switching frequency is low. Therefore, the converter in this paper uses the NLM [5,6].

### 2.2. Design of Sliding Mode Controller

#### 2.2.1. Design of Current Inner Loop Sliding Mode Variable Structure Controller

The dynamic equation of MMC under *dq* coordinate axis is:(1)$$\{\begin{array}{c} L\frac{di_{d}}{dt}+Ri_{d}=u_{d}-v_{d}+\omega Li_{q}\\ L\frac{di_{q}}{dt}+Ri_{q}=u_{q}-v_{q}-\omega Li_{d} \end{array}$$
where $i_{d}$ and $i_{q}$ are the components of the current at the AC side of MMC on *d*-axis and *q*-axis. $u_{d}$ and $u_{q}$ are the components of the access terminal voltage on the AC side of MMC on *d*-axis and *q*-axis. $v_{d}$ and $v_{q}$ are the components of the fundamental wave voltage at the midpoint of the MMC bridge arm on *d* axis and *q* axis.

Equation (1) can be converted into:(2)$$\{\begin{array}{c} {\dot{i}}_{q}=-\frac{R}{L}i_{d}+\omega i_{q}+\frac{1}{L}u_{d}-\frac{1}{L}v_{d}\\ {\dot{i}}_{q}=-\frac{R}{L}i_{q}+\omega i_{d}+\frac{1}{L}u_{q}-\frac{1}{L}v_{q} \end{array}$$

The choice of sliding mode surface and reaching law is the most important part in the design of sliding mode variable controller, which directly affects the control performance of sliding mode variable controller. Considering that integration can eliminate the static error of the system, the integral sliding surface is selected in this paper, as shown in Equation (3) [18,19].
(3)$$\{\begin{array}{l} s_{1}=e_{1}+k_{s}{}_{1}\int_{0}^{t}e_{1}dt\\ s_{2}=e_{2}+k_{s}{}_{2}\int_{0}^{t}e_{2}dt \end{array}$$
where $e_{1}$ and $e_{2}$ are control errors, $e_{1}=i_{d}-i_{dref}$, $e_{2}=i_{q}-i_{qref}$. $i_{dref}$ and $i_{qref}$ are the reference values of the current on *d*-axis and *q*-axis, respectively, $s_{1}$ and $s_{2}$ denote sliding surface, $k_{s1}$ and $k_{s2}$ are sliding surface control parameters, and better steady-state performance can be obtained by adjusting these parameters.

When the system is in steady state, the system status is running on the sliding surface. However, before the system enters the steady state, a control action must be applied to make the system approach the sliding surface.

Select the exponential approach law, as shown in Formula (4):(4)$$\dot{s}=-\varepsilon \mathrm{sgn}(s)-ks\;\;\;\varepsilon >0,k>0$$

s is the switching function, $\dot{s}=-ks$ is the exponential approach term, and its solution is $s=s(0)e^{-kt}$, k is the approaching speed, $\dot{s}=-\varepsilon \mathrm{sgn}(s)$ is the constant velocity approach term, ε is the arrival speed, sgn(s) is a symbolic function, as shown in Formula (5):(5)$$\mathrm{sgn}(s)=\{\begin{array}{cc} 1 & ,s>0\\ 0 & ,s=0\\ -1 & ,s<0 \end{array}$$

Exponential approach law contains both exponential approach term and constant velocity approach term, which enables the system to quickly approach the switching surface.

Introduce ${\left|x\right|}^{2}$ into the constant velocity reaching term of Equation (4) and obtain Equation (6) simultaneously with Equation (3).
(6)$$\{\begin{array}{l} {\dot{s}}_{1}={\dot{i}}_{d}-{\dot{i}}_{dref}+k_{s1}(i_{d}-i_{dref})=-\varepsilon_{1}{\left|x_{1}\right|}^{2}\mathrm{sgn}(s_{1})-k_{1}s_{1}\\ {\dot{s}}_{2}={\dot{i}}_{q}-{\dot{i}}_{qref}+k_{s2}(i_{q}-i_{qref})=-\varepsilon_{2}{\left|x_{2}\right|}^{2}\mathrm{sgn}(s_{2})-k_{2}s_{2} \end{array}$$
where x is the state variable of the system, and the control error is selected in this paper.

At the beginning of control, $-\varepsilon {\left|x\right|}^{2}\mathrm{sgn}(s)$ and −ks worked together, and the approaching speed was fast. As the distance between the state variable x and the sliding surface gradually decreases, both $-\varepsilon {\left|x\right|}^{2}\mathrm{sgn}(s)$ and −ks tend to zero and finally stabilize on the sliding surface. The approach law can adjust the approach speed according to the distance between the state variable and the sliding surface and achieve stable operation while reducing chattering.

According to formula (1), the AC side current $i_{d}$ and $i_{q}$ of MMC converter is affected by control quantity $u_{d}$ and $u_{q}$, grid electromotive force $v_{d}$ and $v_{q}$, and cross coupling term $\omega Li_{d}$ and $\omega Li_{q}$, so feedforward decoupling control is required. By introducing Equation (6) into Equation (2), the sliding mode variable structure control law of the current inner loop of the grid-connected composite device can be obtained, as shown in Equation (7). Figure 3 is the sliding mode variable structure control block diagram of the current inner loop of the GCCD.
(7)$$\left\{ \begin{array}{l} v_{d}=u_{d}+\omega Li_{q}-L\left[k_{1}s_{1}+\varepsilon_{1}{\left|x_{1}\right|}^{2}\mathrm{sgn}(s_{1})+k_{s1}e_{1}\right]\\ v_{q}=u_{q}-\omega Li_{d}-L\left[k_{2}s_{2}+\varepsilon_{2}{\left|x_{2}\right|}^{2}\mathrm{sgn}(s_{2})+k_{s2}e_{2}\right] \end{array}\right.$$

#### 2.2.2. Design of Voltage Outer Loop Sliding Mode Variable Structure Controller

Each bridge arm of MMC is composed of multiple submodules in cascade, and its DC side capacitor voltage is supported by the submodule capacitor voltage. Equation (8) can be obtained from the law of energy conservation [20].
(8)$$\frac{3}{2}(u_{d}i_{d}+u_{q}i_{q})=u_{dc}i_{dc}=C_{eq}u_{dc}\frac{du_{dc}}{dt}$$
where $i_{dc}$ is the current at the DC side of the inverter, and $C_{eq}$ is the equivalent capacitance value at the DC side.

For three-phase balanced power grid, there is $u_{q}=0$, so the DC side dynamic equation of MMC is:(9)$$\frac{du_{dc}}{dt}=\frac{3u_{d}i_{d}}{2C_{eq}u_{dc}}$$

The error between the DC side voltage command value $u_{dcref}$ and the actual voltage $u_{dc}$ is $e_{3}$, $e_{3}=u_{dcref}-u_{dc}$. Considering that the control objective of the voltage outer loop controller is to maintain the stability of the DC side voltage, the design requirements of the controller are to ensure that the DC side voltage control has a strong anti-interference ability, and the differential action can eliminate the influence of disturbance on the system and can better eliminate chattering. Therefore, the first order sliding mode is selected as shown in Formula (10):(10)$$s_{3}=e_{3}+\beta \frac{de_{3}}{dt}$$
where β is the coefficient of differential term.

Substitute Equation (9) into Equation (10) to get Equation (11).
(11)$$\begin{array}{ll} s_{3} & =(u_{dcref}-u_{dc})+\beta ({\dot{u}}_{dcref}-\frac{du_{dc}}{dt})\\ & =(u_{dcref}-u_{dc})+\beta ({\dot{u}}_{dcref}-\frac{3u_{d}i_{d}}{2C_{eq}u_{dc}})\\ & =(u_{dcref}-u_{dc})-\frac{3\beta u_{d}i_{d}}{2C_{eq}u_{dc}} \end{array}$$

Let $s_{3}=0$ get:(12)$$i_{dref}=\frac{2C_{eq}u_{dc}}{3\beta u_{d}}(u_{dcref}-u_{dc})$$

The reference active current $i_{dref}$ of the inner loop controller can be obtained from Equation (12). Figure 4 is the voltage outer loop sliding mode variable structure control block diagram.

### 2.3. Necessity of Parameter Optimization

The most outstanding advantage of the SMC control is that the operation of the system is only affected by the sliding mode surface parameters and is not affected by the original parameters of the system, so that the system has more excellent full adaptability than robustness under certain conditions [18,19].

When the state trajectory of the system reaches the sliding mode surface, it is difficult to slide completely along the preset sliding mode towards the balance point because the actual switching device will have more or less time delay, but it follows the principle of repeated switching on both sides of the switching surface. This is the chattering phenomenon of the system. Chattering affects the control accuracy of the system as well as the service life of devices. However, since chattering is inherent in sliding mode variable control, it cannot be completely eliminated and can only be reduced as much as possible [20]. The magnitude of chattering is directly affected by the parameters of the sliding mode variable controller, so finding the optimal parameters of the sliding mode variable controller can effectively improve the control performance of the system.

## 3. The Moth-Flame Optimization

### 3.1. Algorithm Principle

The *MFO* algorithm is derived from the phenomenon that natural moths approach the light source in a spiral way. The moths are individuals who constantly search for the optimal value, and the flame is the optimal position obtained by the moths in the process of optimization. Each moth seeks optimization around the corresponding flame and updates the flame position when a better solution is found, so as to ensure that the optimal solution is retained during the optimization process [21].

The *MFO* algorithm can be expressed as a triplet optimization problem:(13)$$\{\begin{array}{l} MFO=(I,P,T)\\ I:f\to \{M,OM\}\\ P:M\to M^{\prime }\\ T:M\to \left\{true,false\right\} \end{array}$$
where M is the position of the moth, i.e., the variable to be optimized, OM is the fitness value corresponding to the moth in M, f is the fitness function, P is the spiral position updating mechanism of the moth around the flame, $M^{\prime }$ is the updated position of the moth, T is an iterative judgment function. If it is true, stop the iteration, otherwise continue the iterative optimization.

The P function formula is:(14)$$M_{i}=S(M_{i},F_{j})$$
(15)$$S(M_{i},F_{j})=D_{i}e^{bt}\cos(2\pi t)+F$$
(16)$$D_{i}=\left|F_{j}-M_{i}\right|$$
where t is the displacement variable, which is generated randomly in the interval [−1, 1], b is the shape constant of the helix function, $D_{i}$ is the distance of the ith moth to the flame.

Figure 5 shows the model of moth updating its position around the flame using logarithmic spiral function. For the convenience of analysis, this diagram only shows the model of one moth in one dimension, which can be compared with that of multiple moths and multiple dimensions. $M_{i}$ in the figure is the initial position of the moth, which may fly to the position $M_{1}$, $M_{2}$, $M_{3}$, $M_{4}$, $M_{5}$ when it flies.

If there are n moths in each iteration, the moths will focus on global optimization and affect the accuracy of local optimization, and the value of ranking in the final flame optimization is low. The researchers propose a self-extinguishing mechanism of flame. With the increase in the iteration times of the algorithm, the flame with poor fitness is gradually discarded. Equation (17) is the flame extinguishing formula:(17)$$F_{\min}=round(N-k\times \frac{N-1}{T})$$
where N is the number of species, k is the number of current iterations, T is the total number of iterations.

### 3.2. Improved MFO Algorithm

According to the existing research results, the traditional *MFO* algorithm has the following problems:

(1)The convergence speed of the algorithm is slow in the later period. Considering that the spiral flight search and position update mechanism of the traditional *MFO* algorithm has a certain balance between the global search ability and the local search ability. In the early stage of optimization, the algorithm can quickly approach the relative optimal solution, but after a certain number of iterations, the spiral flight search will limit the moth to a small area. This search method will only make some minor updates to the current position, which will cause the convergence speed of the algorithm to slow down in the later stage.(2)Premature convergence. The *MFO* algorithm does not have a mechanism to jump out of the local optimum. Once it falls into the local optimum, it is difficult to jump out, leading to premature convergence. At the same time, the adaptive flame extinction mechanism of *MFO* algorithm has enhanced the ability of local optimization, but to a certain extent, it reduces the diversity of the population, and will also lead to premature convergence.

In view of the shortcomings of *MFO* algorithm, this paper adopts the following improvement methods.

(1)Good point set initialization [22,23]

The traditional *MFO* algorithm uses random values to set the initial position of the moth, but this method cannot make the initial position of the moth uniformly distributed within the allowable range. The initial position of the moth population can be distributed more uniformly by using the initialization of the GPS initialization instead of the generation mode of the initial position of the moth with random sliding mode and variable parameters.

The construction of the good point is not affected by the space dimension, which can better solve the problem of solving high-dimensional space. Therefore, a relatively good initial population of the moth can be obtained by setting the initial position of the moth with the GPS initialization method. The specific expression of the GPS initialization is shown in Formula (18):(18)$$P_{n}(k)=\left\{\left(\left\{r^{1}k\right\},\ldots ,\left\{r^{i}k\right\},\ldots ,\left\{r^{t}k\right\}\right),k=1,2,\ldots ,n\right\}$$
where $r^{i}=\left\{2\cos\left(2\pi i/p\right)\right\},1\leq i\leq t$, P is the minimum prime number satisfying p≥2t+3.

(2)Path optimization of moth based on the Levy flight strategy [24,25]

This paper combines the Levy flight strategy with the classical *MFO* algorithm. Even if the algorithm is temporarily trapped in the local optimum, it can also jump out of the local optimum through the Levy flight strategy. The probability density function of Levy flight distribution is:(19)$$\mathrm{Levy}\sim u=t^{-\lambda },1<\lambda <3$$
where λ is the power coefficient.

Equation (20) is the Levy flight jump path update mechanism:(20)$$\{\begin{array}{l} s=\frac{\mu }{{\left|v\right|}^{\frac{1}{\beta }}}\\ \mu \sim N(0,\sigma_{\mu }^{2})\\ v\sim N(0,\sigma_{v}^{2})\\ \sigma_{\mu }={\left[\frac{\Gamma (1+\beta )\sin(\frac{\beta \pi }{2})}{\Gamma (\frac{1+\beta }{2})2^{\frac{\beta \pi }{2}}\beta }\right]}^{\frac{1}{\beta }}\\ \sigma_{v}=1 \end{array}$$
where s is the random step size, β=λ−1, μ, v follows normal distribution.

### 3.3. Performance Test of the IMFO Algorithm

To make the test results more comprehensive and objective, six different test functions are selected for performance test.

In Table 2, $f_{1}$~$f_{3}$ is a single peak test function, and $f_{4}$~$f_{6}$ is a multi-peak test function. The IMFO algorithm, the traditional *MFO* algorithm, and the Particle Swarm Optimization algorithm (PSO) three species optimization algorithms are compared for test function simulation to verify the performance of the proposed IMFO. Set all algorithm populations to 30, the maximum number of iterations to 1000, and the dimension to 10.

From the test results in Table 3, it can be clearly seen that the optimization ability of IMFO is significantly better than *MFO* and PSO.

### 3.4. Application of IMFO Algorithm in Parameter Optimization of Sliding Mode Controller

In this paper, the SMC parameters $k_{1}$, $\varepsilon_{1}$, $k_{s1}$, $k_{2}$, $\varepsilon_{2}$, $k_{s2}$, $k_{3}$, $\varepsilon_{3}$, $k_{s3}$, $k_{4}$, $\varepsilon_{4}$, $k_{s4}$ of current inner loop of MMC1 and MMC2 and the SMC parameters β of voltage outer loop of MMC2 are optimized, and the other parameters are set as fixed values. Taking the above parameters as the object of IMFO optimization, Figure 6 shows the flow chart of IMFO algorithm for optimizing the parameters of sliding mode controller of the GCCD.

## 4. Parameter Optimization of Sliding Mode Variable Controller

### 4.1. Objective Function

The ITAE is widely used in the research of controller parameter optimization. This index can better measure the control performance of the controller in complex environments. In the dynamic process, the smaller the ITAE value, the better the control performance of the controller. Considering that taking only the error integral as the objective function may attach too much importance to the integral value and neglect the overshoot, this paper combines the ITAE index with the overshoot, and adds the corresponding weight coefficient as the evaluation index of the sliding mode controller control performance. The modified *CITAE* index is described as:(21)$$J_{CITAE}=\int_{t}^{T}t\left|e(t)\right|dt+\omega \sigma_{p}\%\times 100$$
where t is the time when power transfer occurs, |e(t)| is the absolute value of control error, $\sigma_{p}\%$ is the overshoot, ω is the weight coefficient.

Considering the dynamic characteristics of the GCCD during power transmission, the objective function is set as:(22)$$\begin{array}{ll} Q & =\omega_{DC}\left(\int_{t}^{T}t\left|e_{DC}(t)\right|dt+\omega_{1}\sigma_{DC}\%\times 100\right)\\ & +\omega_{P}\left(\int_{t}^{T}t\left|e_{P}(t)\right|dt+\omega_{2}\sigma_{P}\%\times 100\right)\\ & +\omega_{Q1}\left(\int_{t}^{T}t\left|e_{Q1}(t)\right|dt+\omega_{3}\sigma_{Q1}\%\times 100\right)\\ & +\omega_{Q2}\left(\int_{t}^{T}t\left|e_{Q2}(t)\right|dt+\omega_{4}\sigma_{Q2}\%\times 100\right) \end{array}$$

### 4.2. Python-PSCAD Joint Simulation

The modeling and simulation of the GCCD based on the SMC in this paper is carried out in PSCAD, but the debugging of IMFO algorithm in PSCAD is inefficient and difficult. In order to realize the optimization of sliding mode variable controller parameters by using IMFO algorithm, this paper uses Python-PSCAD joint simulation. Figure 7 is the joint simulation structure diagram.

As shown in Figure 7, the implementation of Python-PSCAD joint simulation should first use Python to achieve IMFO optimization, so as to achieve iterative updating of controller parameters, and assign the updated parameters to the PSCAD simulation model. Then, use Automation Library to control the PSCAD simulation model for simulation operation, and call the MATLAB engine through Python to calculate the fitness value. Finally, the fitness value calculated by the MATLAB engine is read through Python language for the next iteration.

The specific process of parameter optimization of sliding mode variable structure controller is as follows:

Step 1: Import Automation Library controller, open PSCAD, and call MATLAB engine.

Step 2: Positioning and setting the parameter setting device of sliding mode variable structure controller.

Step 3: Use IMFO to optimize the parameters of sliding mode variable structure controller.

Step 4: Simulate the sliding mode variable structure controller parameters obtained in the optimization process on the PSCAD platform to obtain the state information.

Step 5: MATLAB reads the status information and calculates the fitness value.

Step 6: judge whether the iteration stop conditions are met. If they are met, execute Step 7. If not, execute Step 3.

Step 7: Output the minimum fitness value and corresponding control parameters.

## 5. Simulation Verification

### 5.1. Simulation Model Parameters

The simulation model of 41 level the GCCD based on back-to-back MMC-HVDC is built in PSCAD/EMTDC simulation software to verify and analyze the effectiveness of the *MFO* algorithm in the SMC parameter optimization process. The system to be paralleled on both sides is set as the hydraulic turbine model, with the rated capacity of 120 MVA, the generator outlet voltage of 13.8 kV, and the transformer transformation ratio of 13.8/230 kV, the loads of systems on both sides are S1 = 70 + j62 MVA and S2 = 10 + j20 MVA, respectively. The other simulation parameters are shown in Table 4.

This paper takes the comprehensive ITAE value of active power, reactive power and DC side voltage as the evaluation index, and focuses on optimizing the current inner loop sliding mode variable control parameters of MMC1 and MMC2, and the voltage outer loop sliding mode variable control parameters of MMC2. The other parameters are set as fixed values.

The IMFO algorithm proposed in this paper is applied to optimize the control parameters of grid-connected composite devices. Joint operation between Python and PSCAD is realized on a computer with 8 G memory and 2.4 GHz main frequency. The number of algorithm population is set to 30, and the maximum number of iterations is set to 100.

### 5.2. Performance Analysis of Optimization Algorithm

The sliding mode controller parameters of the GCCD are optimized with the traditional *MFO* and the IMFO algorithm and compared with the control performance before optimization. Table 5 shows the evaluation indicators before and after optimization.

It can be seen from the comprehensive ITAE value in Table 5 that *MFO* algorithm and IMFO algorithm can optimize the control parameters to improve the control performance. Moreover, by comparing the optimized comprehensive ITAE values, it can be seen that the optimization effect of the IMFO algorithm is 11.25% higher than that of the *MFO* algorithm. It can be seen that the IMFO algorithm proposed in this paper can better optimize the SMC parameters of the GCCD than that of the *MFO* algorithm.

### 5.3. Parameter Control Effect Comparison after Optimization

It is one-sided to simply verify the control effect with the comprehensive ITAE index. This section compares the parameter control effect before and after optimization with the waveform diagram of the DC side voltage, the active power, the reactive power on the MMC1 side and the reactive power on the MMC2 side.

Figure 8 and Figure 9 show the DC side voltage waveform and its local amplified waveform (1 in the figure represents the local amplification area of t = 3.8~4.0 s). From Figure 8, it can be clearly seen that the DC side voltage fluctuation in the power transmission process without parameter optimization, after *MFO* parameter optimization and after IMFO parameter optimization. Without parameter optimization, the DC side voltage fluctuation under sliding mode control during power transmission is about 0.2 kV, while the DC side voltage fluctuation after *MFO* and IMFO optimization is about 0.04 kV, which shows that the DC side voltage fluctuation amplitude after controller parameter optimization is significantly reduced, and the system stability is improved. In terms of response speed, it takes 0.7 s for the DC side voltage without parameter optimization to return to steady state, while it takes about 0.6 s for the DC side voltage optimized by *MFO* algorithm to return to steady state, while it takes only 0.2 s for the DC side voltage optimized by IMFO algorithm to return to steady state. The above data can effectively show that the DC side voltage response speed after controller parameter optimization is significantly accelerated.

In order to further analyze the influence of parameter optimization on the steady-state performance of the system, the DC side voltage waveform between 3.8 s–4 s is selected as shown in Figure 9. It can be seen that the DC side voltage waveform without parameter optimization and the DC side voltage waveform optimized by *MFO* have a certain steady-state error in the steady-state, while the voltage waveform optimized by IMFO has good steady-state performance without obvious steady-state error. To sum up, the DC side voltage of the system optimized by *MFO* and IMFO parameters has less fluctuation, faster response speed and smaller steady-state error than the DC side voltage of the system not optimized by parameters, and the dynamic and static performance of the system optimized by IMFO parameters is better than that of *MFO*.

Figure 10 shows the active power waveform in the power transfer process. In order to better observe the waveform, it is partially amplified, as shown in Figure 11 (1 in the figure represents the local amplification area of t = 3.03~3.13 s, 2 in the figure represents the local amplification area of t = 3.4~3.5 s). From Figure 11a, it can be clearly seen that the response speed and overshoot of the active power in the power transfer process without parameter optimization, after *MFO* optimization and after IMFO optimization. Without parameter optimization, there is about 0.1 MW overshoot in the active power transfer process, and it takes about 0.14 s for the active power to return to steady state. Figure 11b can reflect the static characteristics of active power. It can be seen that there is no obvious difference in the static performance of active power under the three conditions. To sum up, the active power optimized by *MFO* and IMFO parameters has less overshoot and faster response speed than the active power without parameter optimization, and the dynamic performance optimized by IMFO parameters is better than *MFO*.

Figure 12 shows the reactive power waveform of MMC1 in the power transfer process. In order to better observe the waveform, it is partially amplified, as shown in Figure 13 (1 in the figure represents the local amplification area of t = 3.4~3.5 s, 2 in the figure represents the local amplification area of t = 3.04~3.20 s). From Figure 13a, it can be clearly seen that the response speed and overshoot of the reactive power rate of MMC1 in the power transfer process without parameter optimization, after *MFO* optimization and after IMFO optimization. There is no obvious overshoot in several cases, and it can be seen from Figure 13a that the response speed of reactive power without parameter optimization is about 0.08 s, while the response speed after *MFO* parameter optimization is about 0.13 s. However, according to Figure 13b, the static performance of reactive power without parameter optimization is poor. After the reactive power has returned to steady state, there is still a fluctuation of about 0.01 MVar, while the static performance of reactive power after *MFO* parameter optimization is good. The response speed after IMFO parameter optimization is about 0.07 s, and according to Figure 13b, the static performance of reactive power after IMFO parameter optimization is better than that after *MFO* parameter optimization. In conclusion, the reactive power of the system optimized by IMFO parameters has faster response speed and better static stability than the reactive power of the system not optimized by parameters.

Figure 14 shows the reactive power waveform of MMC2 in the power transfer process. In order to better observe the waveform, it is partially amplified, as shown in Figure 15 (1 in the figure represents the local amplification area of t = 3.04~3.20 s, 2 in the figure represents the local amplification area of t = 3.4~3.5 s). From Figure 15a, it can be clearly seen that the response speed and overshoot of MMC2’s reactive power rate in the power transfer process without parameter optimization, *MFO* optimization and IMFO optimization. Without parameter optimization, there is about 0.05 MVar overshoot in reactive power transfer, Moreover, it takes about 0.14 s for reactive power to return to steady state. However, the overshoot of reactive power optimized by parameters is not obvious, and the reactive power response speed is significantly improved. The reactive power response speed optimized by *MFO* parameters is about 0.13 s, and the reactive power response speed optimized by IMFO parameters is about 0.11 s. Figure 15b can reflect the static characteristics of reactive power. The static performance of reactive power without parameter optimization is poor. After the reactive power has returned to steady state, there is still a fluctuation of about 0.02 MVar. The static performance of reactive power after parameter optimization is good. According to the waveform, the static performance of reactive power after IMFO parameter optimization is better than that after *MFO* parameter optimization. To sum up, reactive power optimized by *MFO* and IMFO parameters has smaller overshoot and faster response speed than reactive power without parameter optimization, and dynamic and static performance optimized by IMFO parameters is better than *MFO*.

To sum up, it can be seen from Figure 8, Figure 9, Figure 10, Figure 11, Figure 12, Figure 13, Figure 14 and Figure 15 that parameter optimization of SMC of the GCCD based on back-to-back MMC-HVDC can further restrain the fluctuation of DC side voltage during power transmission and improve the dynamic and static performance of the system. According to the comparison, compared with the traditional *MFO* algorithm to suppress the DC side voltage fluctuation of the grid-connected composite device and improve the dynamic and static performance of the system, the optimized parameters of the improved *MFO* algorithm have more advantages to suppress the DC side voltage fluctuation of the grid-connected composite device and improve the dynamic and static performance of the system.

## 6. Conclusions

In order to make GCCD controller meet the requirements of different equipment modes and complex working conditions of power grid, this paper proposes to introduce sliding mode control into GCCD controller. Aiming at the difficulty of sliding mode control parameter tuning, a parameter optimization method of GCCD sliding mode controller based on IMFO algorithm is proposed.

(1)This paper establishes the mathematical model of MMC converter, and designs GCCD controller according to SMC principle, so that the GCCD can meet the requirements of different operation modes and complex working conditions of power grid.(2)In this paper, the *MFO* algorithm is improved by using the GPS initialization and the Levy flight strategy, and the IMFO algorithm is proposed. By comparing the data performance of different algorithms under single peak and multi peak standard test functions, it can be seen that the proposed IMFO algorithm can effectively improve the performance of parameter optimization.(3)This paper proposes a sliding mode controller parameter optimization method based on the IMFO algorithm. The *CITAE* index is used as the standard to measure the control performance of sliding mode controller, and the IMFO algorithm is used to optimize the sliding mode controller parameters. Through Automation Library as a link, automatic parameter tuning is realized in Python-PSCAD joint simulation. By comparing the step response performance of non-optimization, *MFO* and IMFO, it can be seen that, compared with the traditional *MFO* algorithm, the IMFO algorithm can effectively reduce the DC side voltage fluctuation of the GCCD, improve the dynamic and static performance of the system, and further improve its control performance.(4)The controller constructed in this paper can effectively improve the dynamic and static performance of the GCCD, but the research background of this paper is the traditional power grid, and the applicability of this controller to the new power system with large-scale new energy access has not been verified. Later, we will continue to study the control strategy of the GCCD for this problem, so that the GCCD can be applied to different scenarios.

## Figures and Tables

**Figure 1 sensors-23-00149-f001:**
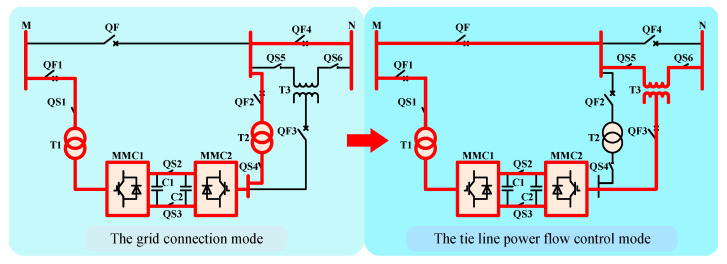
The topology of the GCCD.

**Figure 2 sensors-23-00149-f002:**
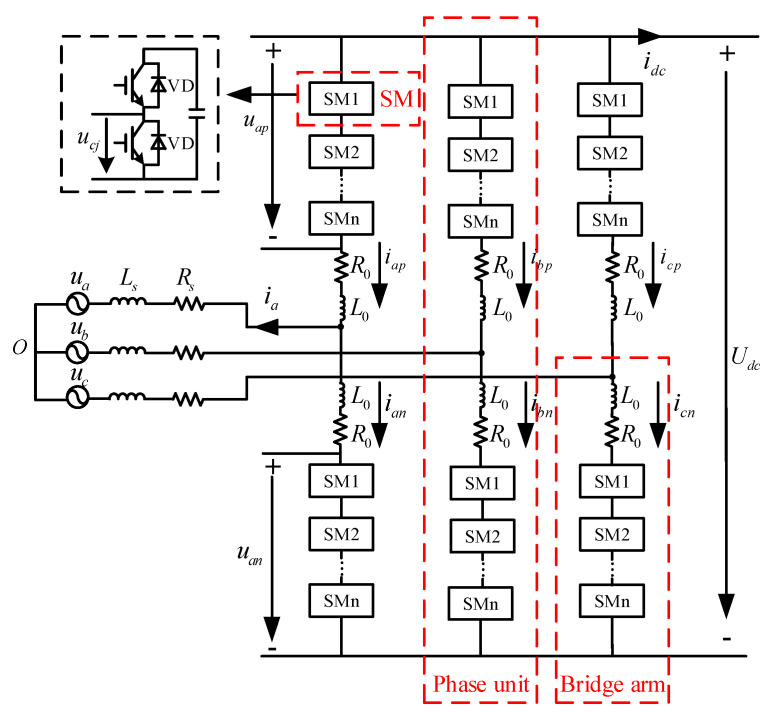
The MMC three-phase topology and sub-module structure.

**Figure 3 sensors-23-00149-f003:**
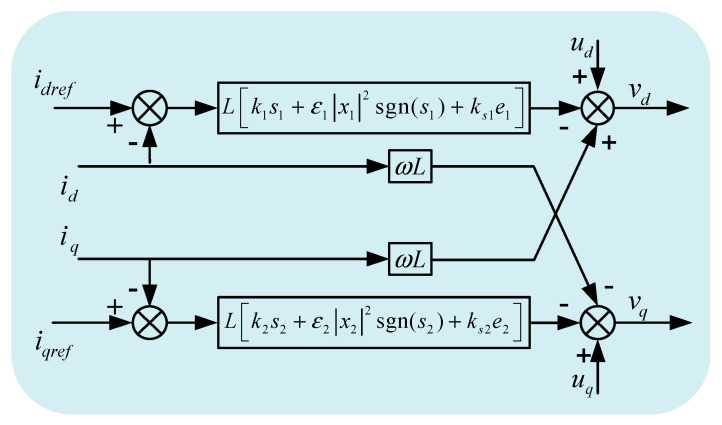
Structural block diagram of current inner loop sliding mode variable control strategy.

**Figure 4 sensors-23-00149-f004:**
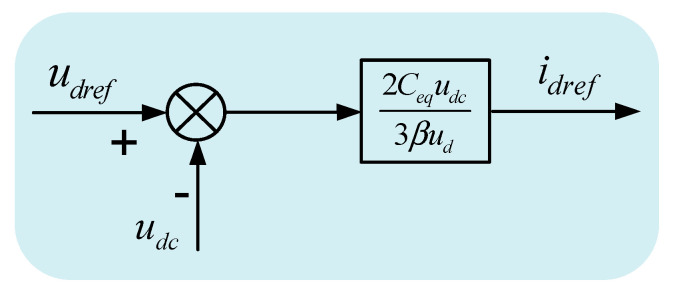
Structural block diagram of current outer loop sliding mode variable control strategy.

**Figure 5 sensors-23-00149-f005:**
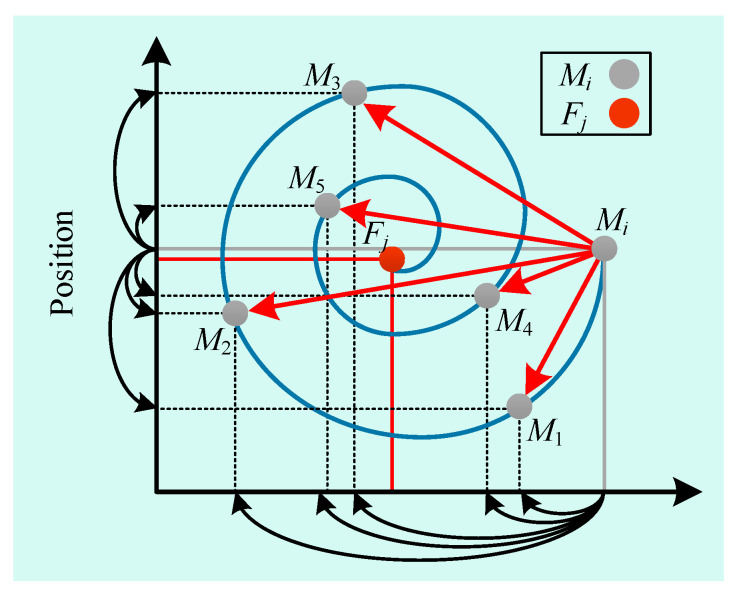
Position updating mechanism of moth.

**Figure 6 sensors-23-00149-f006:**
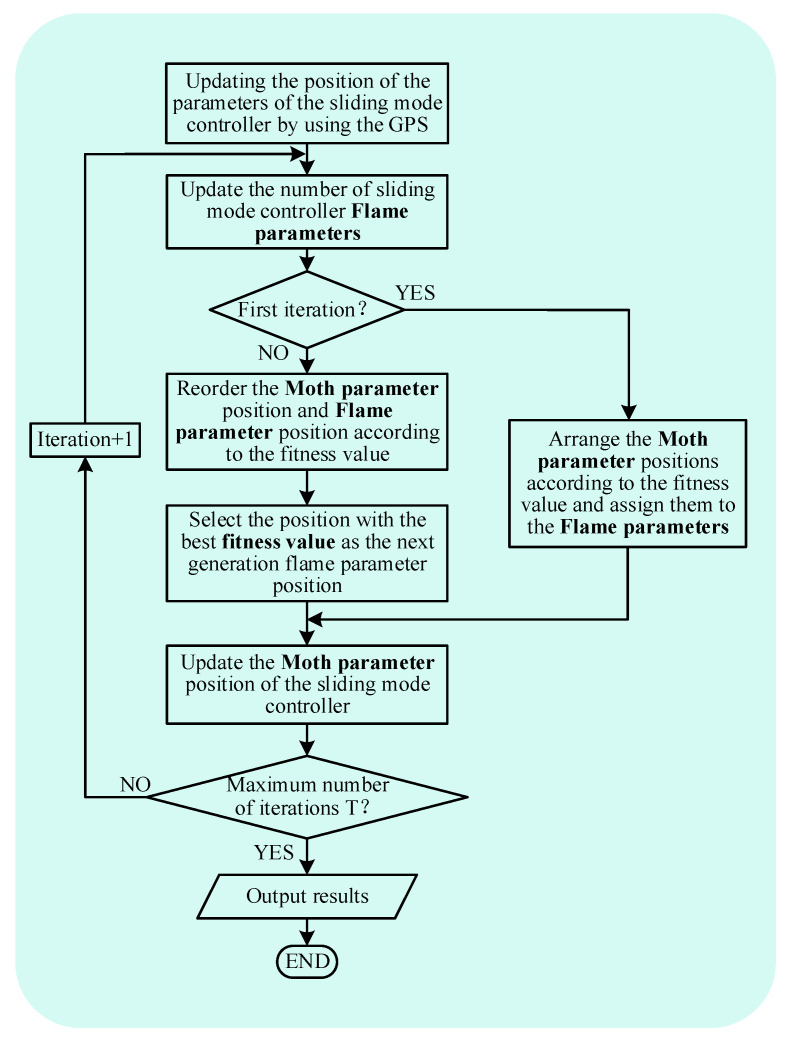
IMFO optimization flow chart.

**Figure 7 sensors-23-00149-f007:**
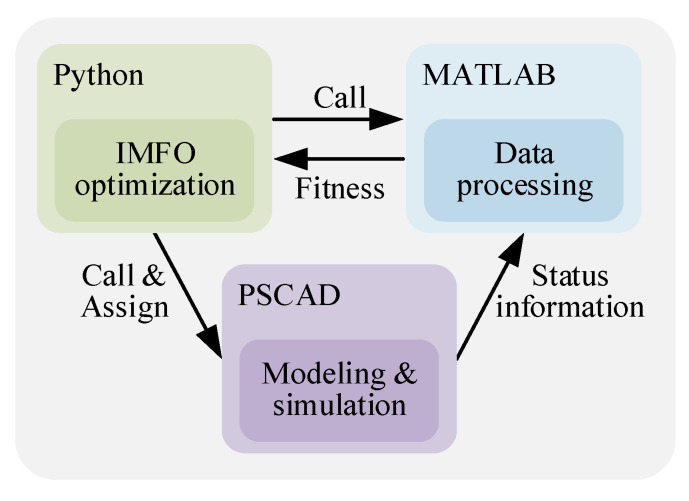
Joint Simulation Structure Diagram.

**Figure 8 sensors-23-00149-f008:**
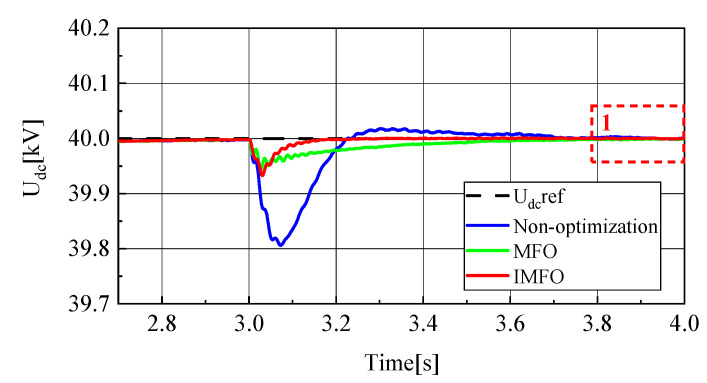
DC side voltage waveform.

**Figure 9 sensors-23-00149-f009:**
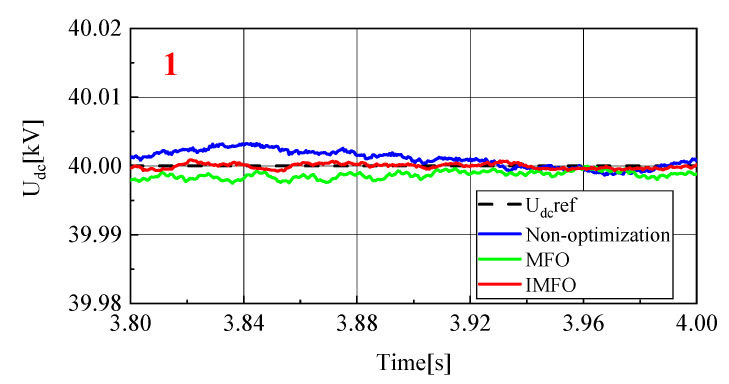
Local amplified waveform of DC side voltage.

**Figure 10 sensors-23-00149-f010:**
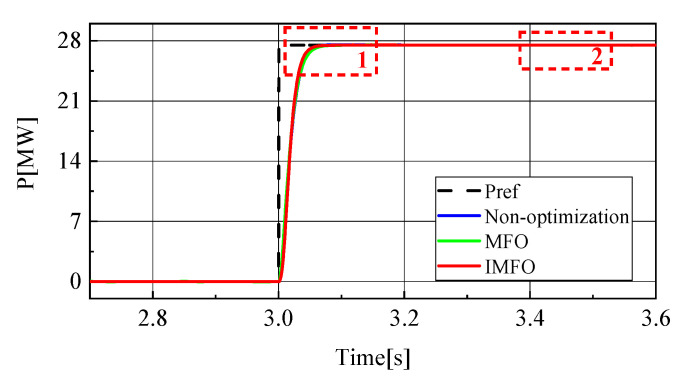
Active power waveform.

**Figure 11 sensors-23-00149-f011:**
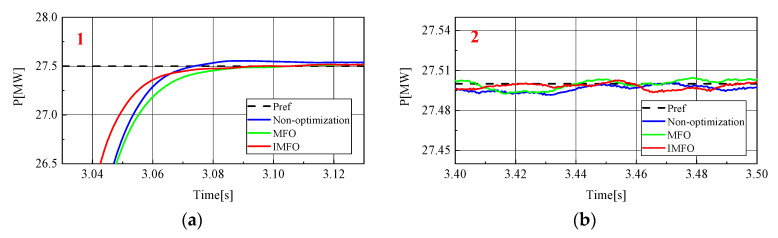
Active power local amplification waveform. (**a**) Locally amplified waveform 1. (**b**) Locally amplified waveform 2.

**Figure 12 sensors-23-00149-f012:**
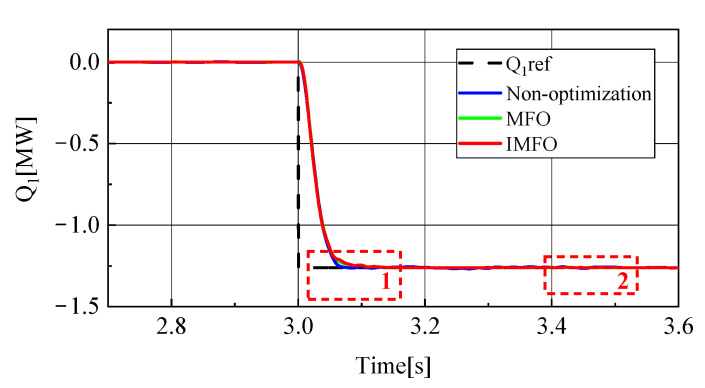
MMC1 reactive power waveform.

**Figure 13 sensors-23-00149-f013:**
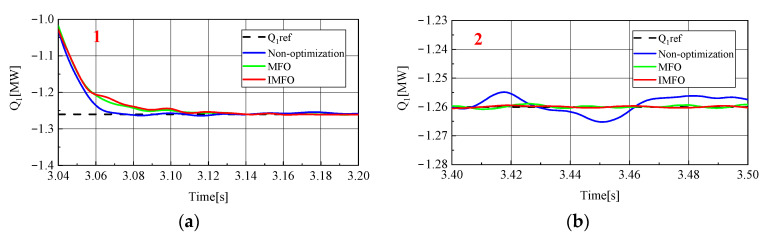
MMC1 reactive power local amplification waveform. (**a**) Locally amplified waveform 1. (**b**) Locally amplified waveform 2.

**Figure 14 sensors-23-00149-f014:**
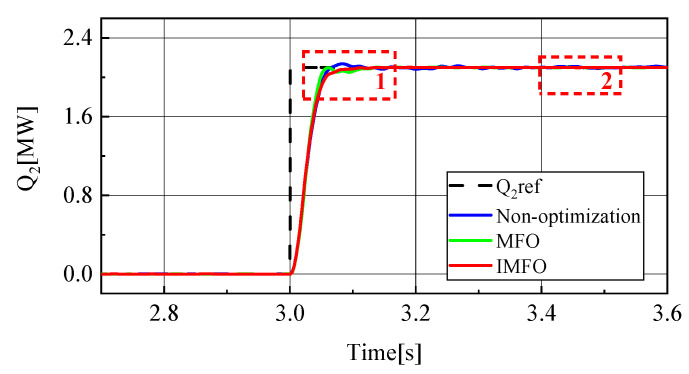
MMC2 reactive power waveform.

**Figure 15 sensors-23-00149-f015:**
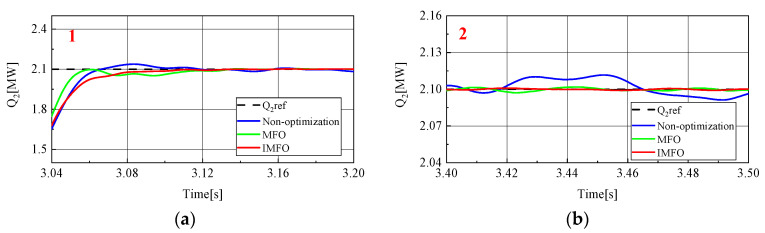
MMC2 reactive power local amplification waveform. (**a**) Locally amplified waveform 1. (**b**) Locally amplified waveform 2.

**Table 1 sensors-23-00149-t001:** Test function.

Control Strategy	Advantage	Disadvantage
PID Control [7]	Simple structure, strong robustness	Proportional control will affect the dynamic performance of the system
Fuzzy control [8]	Strong applicability, strong robustness	The control scheme depends on human experience
Hysteresis Control [9]	Simple parameters, fast response	Frequent switch changes
Sliding mode control [10]	Dynamic performance enhancements	Complex parameters, difficult to set manually

**Table 2 sensors-23-00149-t002:** Test function.

Test Function	Expression	$f_{\min}(x)$
$f_{1}$	$f_{1}(x)=\sum_{i=1}^{n}x_{i}^{2}$	0
$f_{2}$	$f_{2}(x)=\sum_{i=1}^{n}\left|x_{i}\right|+\prod_{i=1}^{n}\left|x_{i}\right|$	0
$f_{3}$	$f_{3}(x)=\sum_{i=1}^{n}{\left(\sum_{j=1}^{i}x_{j}\right)}^{2}$	0
$f_{4}$	$f_{4}(x)=\sum_{i=1}^{n}x_{i}\sin(x_{i})+0.1x_{i}$	0
$f_{5}$	$f_{5}(x)=\sum_{i=1}^{n}\left[x_{i}{}^{2}-10\cos(2\pi x_{i})+10\right]$	0
$f_{6}$	$f_{6}(x)=\frac{1}{4000}\sum_{i=1}^{n}\left(x_{i}^{2}\right)-\prod_{i=1}^{n}\cos\left(\frac{x_{i}}{\sqrt{i}}\right)+1$	0

**Table 3 sensors-23-00149-t003:** Test results of function.

Test Function	Algorithm	Average Value	Variance
$f_{1}$	PSO	2.642245	0.144362
	*MFO*	7.92 × 10^−30^	1.49 × 10^−59^
	IMFO	1.10 × 10^−189^	0
$f_{2}$	PSO	1.09857	0.028251
	*MFO*	1.333333	3.8 × 10^−38^
	IMFO	4.5 × 10^−103^	1 × 10^−208^
$f_{3}$	PSO	19.21238	88.85796
	*MFO*	1.34 × 10^−6^	2.13 × 10^−14^
	IMFO	2.8 × 10^−151^	0
$f_{4}$	PSO	70.70418	0.876751
	*MFO*	12.25906	4.81 × 10^−24^
	IMFO	3.6 × 10^−101^	9.5 × 10^−212^
$f_{5}$	PSO	26.16555	110.6147
	*MFO*	21.62607	24.74814
	IMFO	0	0
$f_{6}$	PSO	1.565486	0.038066
	*MFO*	4.91 × 10^−15^	0
	IMFO	8.88 × 10^−16^	0

**Table 4 sensors-23-00149-t004:** Simulation parameters.

Parameter	Value
system voltage	220 kV
DC side voltage rating	40 kV
MMC capacity	40 MVA
number of bridge arm modules	40
sub module capacitance value	10 mF
inductance value of bridge arm reactor	10 mH

**Table 5 sensors-23-00149-t005:** Comparison of evaluation indexes before and after optimization.

Algorithm	*CITAE* Value
non-optimization	0.442789
*MFO*	0.366769
IMFO	0.358213

## Data Availability

The data presented in this study are available on request from the corresponding author. The data are not publicly available due to the need to use some of the data in this paper in subsequent research.

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
