# Peer review of "Research on Parameter Optimization Method of Sliding Mode Controller for the Grid-Connected Composite Device Based on IMFO Algorithm"

_sensors, 2022, doi:10.3390/s23010149_

Round 1
Reviewer 1 Report
1-What are the novelty and advantages of your work?
2- In the introduction, it is too short and not enough to state the current work. It should be expanded and reconstructed. Including the motivation, the main work, and the improvements compared with previous related works should be emphasized in this section and explain how the present work defers from that published previously.
3- The literature review given in this paper is pore to state the contribution of the present work, as there are recent works that deal with such synchronous machine systems and SMC controller which are not included in this survey such as:
[1] Belkhier, Youcef, et al. "Robust interconnection and damping assignment energy-based control for a permanent magnet synchronous motor using high order sliding mode approach and nonlinear observer." Energy Reports 8 (2022): 1731-1740.
[2] Althobaiti, Ahmed, et al. "Expert knowledge based proportional resonant controller for three phase inverter under abnormal grid conditions." International Journal of Green Energy (2022): 1-17.
Etc…
4- The motivation of the research is not clear and the innovation of the paper is insufficient, if it is not then these should be respectively given.
5- The abstract and introduction is too short and a reader can't get full information of contribution. It must be revised. In particular, the last paragraph of the introduction should be seriously edited.
6- The conclusion is too short as well; it needs to be improved with the main finding.
Author Response
Thank you very much for the comments given by the editor and the timely handling of the manuscript, as well as the suggestions of the reviewers. These comments are very professional and valuable, which help to modify and improve our paper. We have carefully studied all the comments and revised them. Changes to the manuscript are shown in yellow.
We hope the revised version is now suitable for publication and look forward to hearing from you in due course.

Reviewer 2 Report
1. the limitations of the proposed work are missing in the conclusion part
2. major self-citation is reported in the paper that includes references [2-10], which is bad practice in academia. Hence literature review need major modifications
3. include modulation method and topology of MMCs used.
Author Response

(The authors gave the same response as above.)

Round 2
Reviewer 1 Report
The paper can be accepted.